# Learning the Dynamics of Compliant Tool-Environment Interaction for Visuo-Tactile Contact Servoing

**Mark Van der Merwe**     **Dmitry Berenson**     **Nima Fazeli**
Department of Robotics
University of Michigan
{markvdm, dmitryb, nfz}@umich.edu
https://www.mmintlab.com/extrinsic-contact-servoing/

**Abstract:** Many manipulation tasks require the robot to control the contact between a grasped compliant tool and the environment, e.g. scraping a frying pan with a spatula. However, modeling tool-environment interaction is difficult, especially when the tool is compliant, and the robot cannot be expected to have the full geometry and physical properties (e.g., mass, stiffness, and friction) of all the tools it must use. We propose a framework that learns to predict the effects of a robot's actions on the contact between the tool and the environment given visuo-tactile perception. Key to our framework is a novel *contact feature* representation that consists of a binary contact value, the line of contact, and an end-effector wrench. We propose a method to learn the dynamics of these contact features from real world data that does not require predicting the geometry of the compliant tool. We then propose a controller that uses this dynamics model for visuo-tactile contact servoing and show that it is effective at performing scraping tasks with a spatula, even in scenarios where precise contact needs to be made to avoid obstacles.

**Keywords:** Contact-Rich Manipulation, Multi-Modal Dynamics Learning

## 1   Introduction

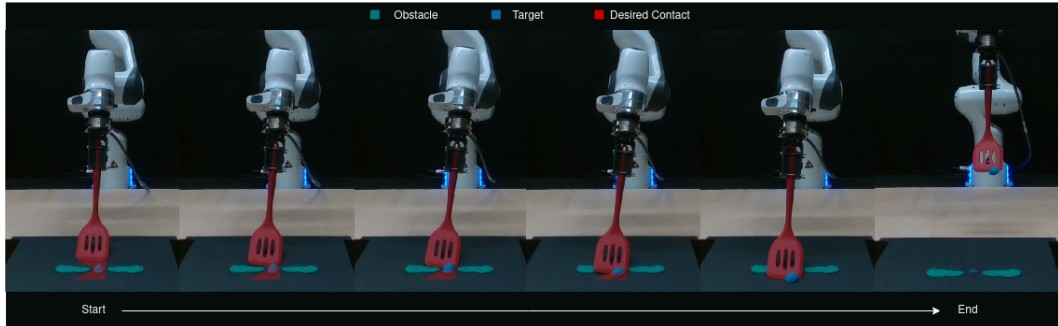

Figure 1: We present a method for *extrinsic contact servoing*, i.e., controlling contact between a compliant tool and the environment. Our method is able to complete the requested contact trajectory, avoiding contact with surface obstacles, and successfully scrape the target object. Note that to do this the spatula must be tilted so that only a corner of it is in contact.

Many manipulation tasks require the robot to control the contact between a grasped tool and the environment. The ability to reason over and control this *extrinsic* contact is crucial to enabling helpful robots that can scrape a frying pan with a spatula, erase or wipe a surface [1], screw a bottle cap onto a bottle [2], perform peg-in-hole assemblies [3, 4], and perform many other tasks.

6th Conference on Robot Learning (CoRL 2022), Auckland, New Zealand.

In this work, we seek to address the problem of controlling the extrinsic contact between a grasped *compliant* tool (e.g. a spatula) and the environment. In general, the robot cannot expect to have the full geometry and physical properties (e.g., mass, friction, stiffness) of all the tools it must use or the geometries of the environments it must manipulate in. Instead, the robot must utilize multimodal sensory observations, such as pointclouds and tactile feedback, to act on the environment.

In recent years, learning-based methods have become increasingly popular to address the complexities of robotic manipulation, including for contact-rich tasks [5]. These methods can be loosely grouped into model-free methods, that directly learn a policy [3, 2, 6], and model-based methods, that learn system dynamics [7, 8, 9]. By focusing on modeling system dynamics, model-based methods can plan to reach new goals without retraining, and are often more data-efficient [9]. Therefore, we propose learning the dynamics of our system to solve the extrinsic contact servoing task.

It is not obvious which representation to use for these dynamics. Fully recovering tool and environment geometries from visual data [10, 11] and tactile feedback [12] has been widely explored, with recent extensions to compliant geometries [13]; however, even if the system can be fully identified, contact models to resolve interactions can have limited fidelity [14]. On the other hand, learned dynamics representations can be difficult to interpret and require demonstrations or observations from the desired state to specify goals [7, 15]. Instead, we propose a novel *contact feature representation* for our learning method that focuses on tool-environment interaction and bypasses explicitly modeling the whole system. We represent the contact configuration as 1) a **binary contact mode** (indicating if the system is in contact); 2) a **contact geometry** (as a line in 3D space); and 3) an **end-effector wrench**.

We propose a learning architecture to model the dynamics of the proposed contact representation from raw sensory observations over candidate action trajectories. We propose structuring the model as a latent space dynamics model with a decoder that recovers the contact state. We also propose an action offset term in the dynamics that allows us to accurately propagate robot poses, despite controller errors (e.g. from robot impedance). To provide labels to our model, we collect self-supervised data on a 7DoF Franka Emika Panda, using sensor data to automatically label contact state.

We validate our proposed method by completing various desired contact trajectories on the real robot system. We first show that our method can track diverse desired contact trajectories in the absence of obstacles. Next, we demonstrate that we can utilize extrinsic contact servoing to scrape a target object from the table, while handling occlusions and avoiding contact with obstacles (Fig. 1).

## 2    Related Work

Existing research has investigated the task of recovering contact locations. Manuelli et al. [16] localize point contacts on a rigid robot with known geometry by employing a particle filtering approach to update a set of candidate contact locations based on force torque sensing. Kim et al. [4] and Ma et al. [17] model contact between a grasped rigid object and the environment by assuming stationary line contacts and modeling the deformation of a GelSlim gripper. The estimated line contact is then used in a Reinforcement Learning (RL) policy. Neither of these methods extends to compliant tools and neither models the dynamics of the contact configuration.

Other works explore *tactile servoing* methods, where contact at the sensor is driven to a desired configuration. Li et al. [18] use a large tactile pad and define contact configuration features of objects pressed against the sensor. They manually construct a feedback controller based on these features and use it to drive contacts to desired configurations. Sutanto et al. [19] use a smaller profile tactile sensor and learn the dynamics of a learned latent space. They then employ a Model Predictive Control (MPC) scheme to drive contacts to desired configurations on the sensor. Both of these works assume contact is happening at the sensing location. We, on the other hand, seek to servo *extrinsic* contacts, where we do not get direct sensing at the point of contact.

Other work focuses on maintaining contact between a tool and the environment. Sakaino [20] uses imitation learning to learn a controller able to maintain contact between a mop and a tabletop. In contrast, we wish to not only maintain contact but control the extrinsic contact geometry.

## 3 Problem Formulation

We parameterize our contact feature as a binary contact indicator $c^b \in \{0, 1\}$, used to indicate whether the tool is in contact, a contact line $c^l \in \mathbb{R}^{2 \times 3}$ representing the contact geometry between the tool and the environment, and an end effector wrench $c^w \in \mathbb{R}^6$. The geometry $c^l$ is only active when the tool is in contact $c^b = 1$. The contact representation allows extrinsic contact goals to be expressed as *desired contact trajectories* $G = [g_1, g_2, \dots, g_L]$, where each $g_i \in \mathbb{R}^{2 \times 3}$ is a desired contact line to reach. We assume that contact should be maintained throughout the task.

We formulate extrinsic contact servoing as a model predictive planning problem, given observations of the current state of the system $o_0$. For a given horizon $T$, we select the next $T$ desired contact lines $[g_{i+1}, \dots, g_{i+T}] \subseteq G$ to be our current contact goal sequence. The planning problem is:

$$
\min_{\boldsymbol{a}_{0:T-1}} \sum_{t=1}^{T} d(\boldsymbol{c}_t^l, \boldsymbol{g}_{i+t})
$$
$$
\text{s.t. } c_t^b = 1, \forall t \in [1, T] \tag{1}
$$
$$
\{c_{0:T}^b, \boldsymbol{c}_{0:T}^l, \boldsymbol{c}_{0:T}^w\} = g(\boldsymbol{o}_0, \boldsymbol{a}_{0:T-1})
$$

Here $g$ is a model describing the *contact feature dynamics*. The binary constraint ensures that the tool remains in contact while the cost function $d$ measures the distance between the two contact lines, as the average Euclidean distance between the line endpoints. Finally, if $d(\boldsymbol{c}_1^l, \boldsymbol{g}_{i+1}) < \epsilon$ we increment $i$, thus moving to the next sequence of desired contact lines for the next round of planning.

## 4 Method

### 4.1 Contact Feature Dynamics Model

To solve our constrained optimization Eq. 1, we require a model $g$ which can map from raw observations $o_0$ and a proposed action trajectory $\boldsymbol{a}_{0:T-1}$ to the resulting contact states $\{c_{0:T}^b, \boldsymbol{c}_{0:T}^l, \boldsymbol{c}_{0:T}^w\}$. We propose modeling the contact feature dynamics as a deep neural network. Our actions are changes in end effector pose.

We assume access to a pointcloud $\boldsymbol{v}_0$ and input wrench $\boldsymbol{h}_0$ measured at the robot's wrist as our observations, $\boldsymbol{o}_0 = (\boldsymbol{v}_0, \boldsymbol{h}_0)$. Note that end effector wrench is both an input to our method and part of the contact state; predicting future wrench aids the representation learning and provides expected wrenches for planning.

We perform all learning in the local end effector frame. We transform the pointcloud to the end effector frame ${}^{EE_0}\boldsymbol{v}_0$ and clip to a $0.5m^3$ bounding box region around the end effector that contains the contact event. We similarly predict our contact lines in the current end effector frame, ${}^{EE_t}\boldsymbol{c}_t^l, \forall t \in [1, T]$. Learning in the end effector frame provides invariance in the visual domain to translations and rotations of the end effector and removes distractors that do not contribute to the contact state, such as the rest of the robot arm or the scene background.

Our contact feature dynamics model (Figure 2) has three components: an encoder $e$ which maps from raw observations to a learned latent space, a decoder $d$ which maps from the latent space to the contact state, and a dynamics model $f$ which captures dynamics in the latent space. We parameterize the models by a set of learned weights $\theta$.

We start by embedding the current observations into the latent space with our encoder $\hat{z}_0 = e(\boldsymbol{v}_0, \boldsymbol{h}_0)$. We unroll actions in the latent space as the contact state alone has insufficient contextual information (e.g. end-effector pose and local geometry information) to predict the next contact state. Because we predict the $t$th contact state in the current end effector frame $EE_t$, an important consideration when designing our dynamics model is being able to accurately recover this frame. Controller error, e.g., from the impedance of the robot, means the commanded action is not perfectly executed. To account for this, we predict an additional term from our dynamics model $\Delta \hat{\boldsymbol{a}}_{t+1}$, which is an $SE(3)$ transformation that predicts the offset between the commanded and realized next end effector pose. Thus, our dynamics model predicts, $\hat{z}_{t+1}, \Delta \hat{\boldsymbol{a}}_{t+1} = f(\hat{z}_t, \boldsymbol{a}_t)$. This allows us to construct the following recursive estimate of our end effector frame ${}^{W}\hat{T}_{EE_{t+1}} = {}^{W}\hat{T}_{EE_t} T(\boldsymbol{a}_t) T(\Delta \hat{\boldsymbol{a}}_{t+1})$, where ${}^{W}\hat{T}_{EE_t}$ is the $SE(3)$ transformation describing the pose of the end effector at time $t$. We know the initial transform ${}^{W}\hat{T}_{EE_0}$ from our robot proprioception, $T(\boldsymbol{a}_t)$ pro-

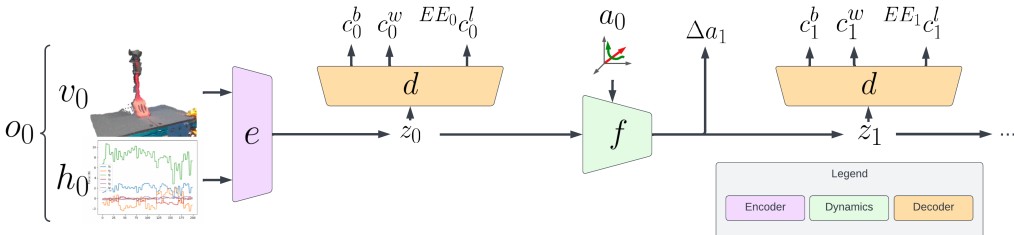

Figure 2: Our proposed contact feature dynamics model. Our architecture embeds raw observations into a latent space where dynamics can be unrolled. We then decode the contact state from the latent space. We also predict an action offset term in order to accurately predict future robot poses.

vides the transformation for the action command, and $T(\Delta \hat{a}_{t+1})$ for the predicted offset term. To enforce valid $SE(3)$ predictions we predict rotations in the axis-angle representation [21].

Finally, we recover our contact state estimates with our decoder $d$ given the latent state $\hat{z}_t$: $\hat{c}_t^b, {}^{EE_t}\hat{c}_t^l, \hat{c}_t^w = d(\hat{z}_t)$. We can then recover the predicted contact line in the world frame by composing with the estimate of the end effector frame transformation ${}^W \hat{T}_{EE_t}$. An overview of the model architectures is shown in Fig. 2 and full architecture details can be found in Appendix A.

#### 4.1.1 Training Loss

We train our model on rollouts of the system, where a single example is a sequence $[\boldsymbol{v}_t, \boldsymbol{h}_t, \boldsymbol{a}_t, \Delta \boldsymbol{a}_t, \boldsymbol{c}_t^l, \boldsymbol{c}_t^w, c_t^b]_{t=0}^T$. We define the loss over the example as:

$$
\begin{aligned}
\mathcal{L}_\theta =& (\sum_{t=0}^T BCE(\hat{c}_t^b, c_t^b) + \alpha \cdot c_t^b \cdot MSE(\hat{\boldsymbol{c}}_t^l, \boldsymbol{c}_t^l) + \beta \cdot MSE(\hat{\boldsymbol{c}}_t^w, \boldsymbol{c}_t^w)) \\
&+ (\sum_{t=1}^T \rho \cdot MSE(\Delta \hat{\boldsymbol{a}}_t, \Delta \boldsymbol{a}_t) + \gamma \cdot MSE(\hat{\boldsymbol{z}}_t, e(\boldsymbol{v}_t, \boldsymbol{h}_t)))
\end{aligned}
\tag{2}
$$

Here BCE is the Binary Cross Entropy classification loss and MSE is the Mean Square Error regression loss. $\alpha, \beta, \rho$ and $\gamma$ are loss weighting terms. The first four loss terms are prediction losses over the contact mode, contact geometry, end effector wrench, and action offset transformation. The final loss term is a latent consistency loss, which encourages latent rollouts to match the latent state yielded by encoding future observations.

### 4.2 Extrinsic Contact Servoing Controller

We propose to solve our planning problem using Model Predictive Path Integral (MPPI), which has been shown to be effective for continuous control tasks where sampling is cheap and parallelizable (e.g., neural network representations) [22]. We convert our binary constraint to a penalty, penalizing a trajectory if it yields actions that lead out of contact. Pairing this with the contact line prediction loss yields the following final cost function:

$$
\sum_{t=1}^T d({}^W \hat{\boldsymbol{c}}_t^l, \boldsymbol{g}_{i+t}) + \phi \cdot \begin{cases} |\hat{c}_t^b - \psi| & \text{if } \hat{c}_t^b < \psi \\ 0 & \text{o.w.} \end{cases}
\tag{3}
$$

The constraint is violated if the likelihood of binary contact is below the classification threshold $\psi$, in which case we penalize by the distance to the threshold. $\phi$ weights the penalty against the contact line loss. With this cost function we apply MPPI to yield the next action and execute it on the robot.

### 4.3 Extrinsic Contact Dynamics Labeling

Our contact dynamics training loss in Eq. 2 requires ground truth contact state labels $(\boldsymbol{c}_t^l, c_t^b, \boldsymbol{c}_t^w)$ at time $t$. As accurate simulation of contactful interactions is challenging, we propose a method of data acquisition directly in the real world. To generate contact line labels, we use a high resolution, low frequency scanner, a Photoneo PhoXi 3D Scanner, to generate high quality scans of the contact interaction. Using these scans, we generate contact line labels by filtering points just above the table,

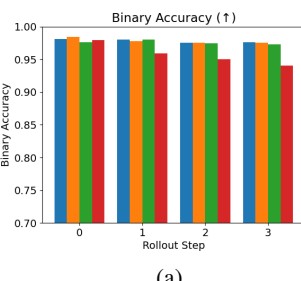 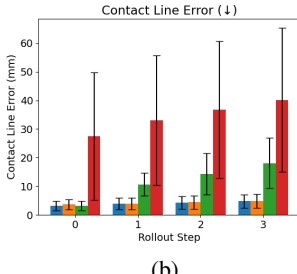 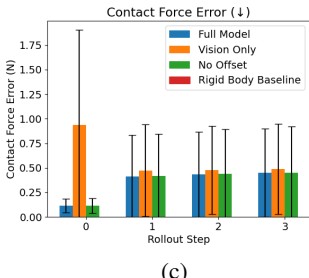

(a)            (b)            (c)

Figure 3: Contact Feature Dynamics Performance: our results show the importance of modeling the action offsets and the compliance of the tool; without both, contact line estimates drift. The Rigid Body Baseline does not predict contact wrenches, so is not shown in (c).

clipped to the area around the end effector. We then cluster these points to remove noisy points on the tabletop and generate the contact line $c_t^l$ by selecting the two furthest points in the cluster. See Appendix B for examples of contact labels. We use a force torque sensor to identify the contact state wrench $c_t^w$. We identify binary contact $c_t^b$ automatically from whether a line was found in the pointcloud and/or based on force torque sensing.

## 5 Results

### 5.1 Experimental Setup

We test our method on a Franka Emika Panda with rigidly-mounted compliant spatulas at the end effector (Fig. 1). For our observations $o_0$, we use pointclouds $v_0$ from an Intel Realsense D435 sensor[1] and mount an ATI Gamma Force/Torque sensor between the end effector and tool. We use the last four wrench values received after the previous action completed as the tactile input $h_0$. To collect our datasets, we use a random action policy with a heuristic to encourage contact between the tool and the spatula. No other objects are on the tabletop during data collection to allow proper data supervision, as detailed in Sec. 4.3.

### 5.2 Baseline

We compare our proposed method to modeling the contact dynamics as a rigid system. We assume the commanded actions are perfectly executed by the robot to recover the future poses of the end effector. We assume access to the tool geometry as a pointcloud in the end effector frame. This pointcloud is then transformed via the future poses of the end effector to recover where the tool would be, assuming rigid motions. We further assume that we know the table location and identify any points in the transformed point cloud that penetrate the table surface. If any exists, we set $c^b = 1$. We then choose the two furthest points in the intersecting set of points as the end points of the contact line $c^l$. The baseline does not predict the wrench $c^w$, but is enough to solve our planning problem in Eq. 1. This baseline makes three assumptions our method does not make: 1) it assumes access to a pointcloud of each tool, 2) it assumes knowledge of the current tool being used, 3) it assumes explicit knowledge of the environment.

### 5.3 Modeling Contact Feature Dynamics

We first investigate the ability of our model to capture the contact feature dynamics exhibited in our dataset. We train three variations on our model. First is the full model, as described in Sec. 4.1, hereafter called "Full Model." Second, to understand the importance of modeling the action offset of the robot, we ablate the offset action prediction, thus we propagate the end effector frame only with the commanded action. We call this method "No Offset." Finally, we investigate our model trained only on visual input data, called "Vision-Only."

We train on a dataset collected from three spatulas (see "Training Tools" in Fig. 5a), collecting 200 trajectories on each, for a total of 30000 transitions. We split the data 80/10/10 for train, validation, and test. We train with a rollout horizon of $T = 3$. All methods are trained with the Adam opti-

---

[1]We don't use the high-fidelity Photoneo scan as it is a very low-frequency scanner.

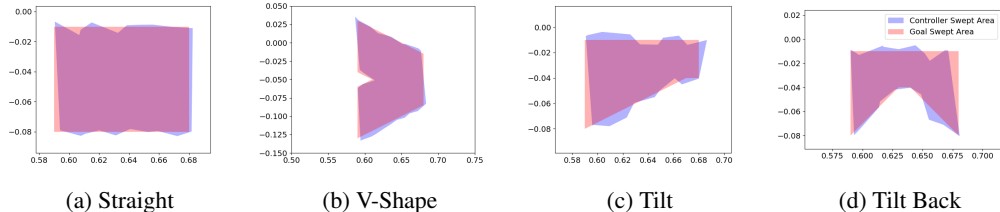

|         |         |         |            |
|:-------:|:-------:|:-------:|:----------:|
| (a) Straight | (b) V-Shape | (c) Tilt | (d) Tilt Back |

Figure 4: Qualitative Scraped Areas: our "Full Model" controller (blue) is able to closely match desired contact trajectories (red).

mizer [23] until convergence on the validation set. We set $\alpha = 100.0, \beta = \gamma = \rho = 0.1$ in our loss term in Eq. 2 to balance the scale of the terms.

We compare the prediction performance of the models on the test split of the dataset in Fig. 3 (see Appendix C for torque prediction results, whose results are very similar to force results). Our full model achieves high accuracy in predicting binary contact ($> 95\%$), 3-5mm contact line error, and less than 0.5N force error. Comparing "Full Model" to "No Offset" shows the importance of modeling action offsets, without which contact line error quickly grows. Comparing all learned models to the Rigid Body Baseline (Sec. 5.2) shows the importance of modeling the compliance of the tool. The "Vision-Only" model unsurprisingly struggles to recover contact forces. Appendix C show additional results examining generalization of the model to the dynamics of an unseen spatula.

### 5.4 Extrinsic Contact Servoing

Next, we investigate how our proposed controller performs following specified contact trajectories.

**Obstacle-Free:** We start by attempting to servo along four different contact trajectories in the obstacle-free environment. The desired contact trajectories are shown in Fig. 4, and explore translation of contact as well as cases where the robot must tilt the tool to achieve a contact smaller than the width of the tool. We use the same labeling technique introduced in Sec. 4.3 to get ground truth contact trajectories executed by the controller. We run the experiment once on a training spatula (left-most in Fig. 5a) and once on an unseen spatula (right-most in Fig. 5a).

We use the controller described in Sec. 4.2, with $\psi = 0.45, \phi = 0.05$ and compare our "Full Model" learned dynamics (as trained in Sec. 5.3) vs the "Rigid Body Baseline" dynamics. To investigate the planning performance, we run the controller five times per trajectory and measure the Intersection over Union (IoU) of the desired and swept contact areas. We construct the goal contact area by sweeping the space between the specified goal contact lines and the realized controller swept area by assuming that the space between two consecutive contact states was swept out if the two states were both in contact.

We show qualitative examples of the "Full Model" controller realized scrapes on the training spatula compared to the goal scrapes in Fig. 4. We see that the controller is able to closely match the desired swept areas, including in the difficult tilting problems.

The IoU performance is shown for the training spatula (Fig. 5b) and unseen spatula (Fig. 5c). Our proposed method outperforms the baseline on nearly all cases, for both the training and unseen tool runs. Performance drops for all methods on the tilting problems, as it is more difficult to maintain dexterous contact on only a part of the tool.

**With Obstacles:** We next examine our method's robustness to visual occlusions and reaction forces arising from contact with a target object to be scraped. This task is common in construction, cooking, and cleaning. A deformable and slightly adhesive material (Playdough) is pressed onto the surface and a contact trajectory is specified through the object. In one case, the target object is alone on the tabletop (Fig. 6), and thus we specify a contact trajectory using the full width of the tool. In the second scenario, obstacles are on the table near the target object (Fig. 1), thus we must specify a contact trajectory that avoids them. All experiments are performed on the leftmost spatula in Fig. 5a. We use our "Full Model", and train on 22005 sequences collected only with the relevant tool.

Besides running our Full Model in these scenarios, we investigated enhancements of the method to aid its performance in the presence of visual occlusions and object reaction forces. First, we

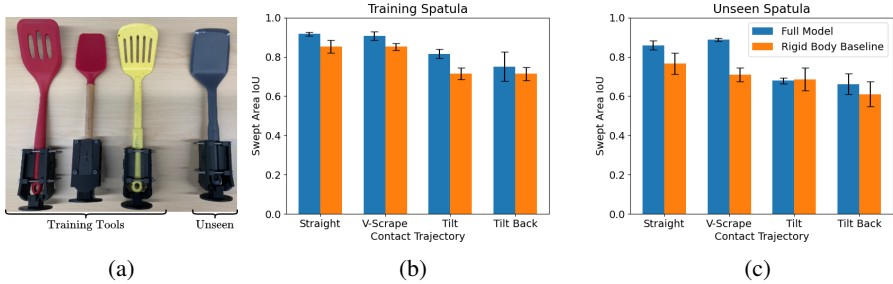

(a)            (b)            (c)

Figure 5: (a) Tools used for experiments. (b),(c) Extrinsic contact servoing IoU performance on a training spatula (b) and unseen spatula (c). Our proposed method tracks the desired contacts with higher IoU, compared to a rigid body baseline method, even when running on an unseen tool.

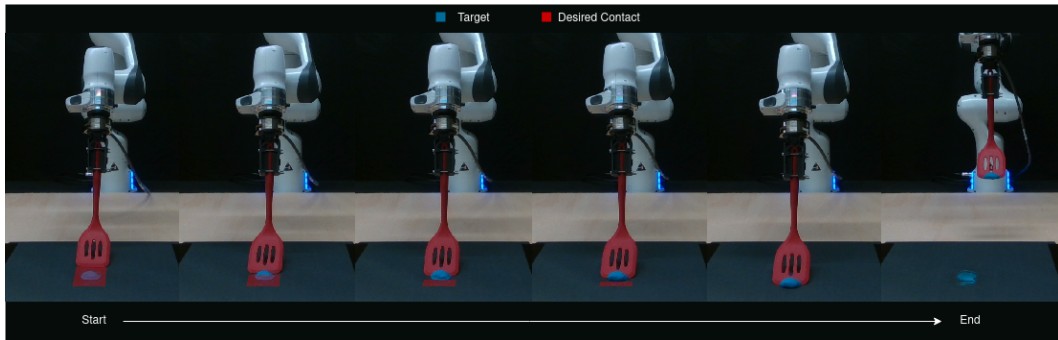

Figure 6: Example of extrinsic contact servoing execution for the "Straight" target obstacle scrape experiment. Our method is able to accurately servo along the desired contact trajectory and successfully scrape the target.

applied data augmentation to our training dataset, randomly generating ellipsoids in the pointcloud and using a hidden point removal algorithm [24] to provide corresponding occlusions in the original point cloud. See Appendix B for examples of augmented inputs. We call this method "Full Model + Aug." Second, we investigate using the difference between the predicted and observed wrenches $\Delta \boldsymbol{w} = \hat{\boldsymbol{c}}_t^w - \boldsymbol{c}_t^w$ to derive an action offset to compensate for the extra wrench experienced by the robot. From $\Delta w$ we derive an action that will counteract this wrench offset $\hat{\boldsymbol{a}}_t = \frac{\Delta \boldsymbol{w}}{\boldsymbol{k}_p}$. $\boldsymbol{k}_p$ is the pose gain of the impedance controller. The offset action is composed with the original action from the controller. We call this method "Full Model + Aug + Wrench Offset."

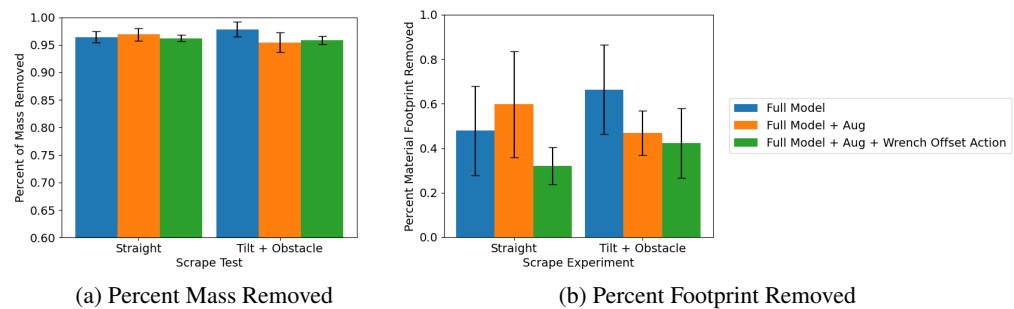

(a) Percent Mass Removed            (b) Percent Footprint Removed

Figure 7: Target Scraping Results: Our "Full Model" and variations for addressing visual occlusions and reaction forces arising from contact with the target object perform comparably on both metrics over 5 trials on each experiment.

We use two metrics. First, we measure the approximate mass of the target object to be scraped before and after scraping and determine the percentage of mass successfully removed. Second, we compare the 2D footprint of the material before and after scraping and report the percentage of the footprint successfully removed. The second is a more challenging metric, since even a slightly wrong scrape will leave residue. The quantitative results over 5 runs of each method in each experiment setup are shown in Fig. 7. Examples of scrape executions are shown in Fig. 1 and Fig. 6. See Appendix C for more examples of scrape results.

We see that in each case, all methods were able to remove over 95% of material mass on average and about 40-60% of material's footprint from the tabletop. Surprisingly, we don't see a consistent improvement training our model with visual occlusions or adding action offsets. The Full Model's robustness to occlusions here could be due to the fact that we use a single tool in these experiments, and thus it may be sufficient in most cases to capture the location of the table with respect to the end effector in order to estimate the contact line. Even with visual occlusions near the tool contact, it is likely our method can still recover the relative pose of the tabletop from the surrounding points. The lack of clear improvement from the wrench offset action may be due to the fact that it is sufficient to be able to replan, as we do at every step with our MPPI controller.

## 6 Limitations and Conclusion

**Limitations:** A common failure mode for our method is in controlling contact when the tool is tilted, where it is more likely for the method to yield actions that take the robot out of contact. This could, in part, be due to data imbalance. In future work, we are interested in utilizing online learning [9] or curiosity [25] to more effectively cover the space of contacts in our dataset.

There are cases where tool to environment contact is not represented well as a contact line. Extending our contact feature dynamics to these tasks will require expanding our representation learning method to consider these more diverse contact specifications and more complex contact modes.

Finally, our method relies upon supervision. For future contact-rich tasks of interest, the need for labels could become more costly. We hope to investigate how we can remove reliance upon supervision by exploring few/zero shot generalization [26, 27] and domain randomization techniques [28].

**Conclusion:** Our approach simplifies contact rich interactions for compliant tool manipulation, by avoiding the necessity for full system identification while maintaining interpretability and accuracy by explicitly modeling the *contact state* of the system and how it evolves. In the future, we wish to investigate our method's applicability to other tasks where full state estimation is difficult, but the contact state is crucial, such as wiping with a cloth. Additionally, we wish to investigate representations that handle more complex contact geometries.

**Acknowledgments**

This material is based upon work supported by the National Science Foundation Graduate Research Fellowship Program under Grant No. 1841052. Any opinions, findings, and conclusions or recommendations expressed in this material are those of the authors and do not necessarily reflect the views of the National Science Foundation. This work was supported in part by Toyota Research Institute under the University Research program 2.0. This work was supported in part by ONR grant N00014-21-1-2118 and NSF grants IIS-1750489 and IIS-2113401.

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
