# OpenReview forum: "Learning the Dynamics of Compliant Tool-Environment Interaction for Visuo-Tactile Contact Servoing"
_robot-learning.org/CoRL/2022/Conference — CoRL 2022 Poster_

### Official Review · Reviewer_DDzt · 2022-07-24

**Originality:** Very Good
**Technical Quality:** Very Good
**Clarity Of Presentation:** Very Good
**Impact:** 3

**Recommendation:**

Weak Accept: I recommend accepting the paper, but will not argue for my recommendation if the majority of other reviewers have a different opinion.

**Summary:**

The paper propose a method to learn the dynamics of these contact features from real world data with unknow tool geometry, and propose a controller that uses learned dynamics model for visuo-tactile contact servoing and show that it is effective at performing scraping tasks with a spatula.

**Issues:**

1. Format of formulas (1), (2), (3)  and (4), (5) are wrong. The formulas (4),(5) are only one, not two.
2. For line contacts model, different geometry and physical properties (e.g., mass, stiffness, and friction) tools are needed to proof the generalization of the framework.
3. How about the performance on discontinuous surface, such as v-shape surface?
4. How about the performance of rotating the spatula comparing to pushing?

**Quality Of The Limitations Section:**

Limitations are addressed clearly

**Reviewer Expertise:**

4: The reviewer is confident but not absolutely certain that the evaluation is correct

**Robotics Focus:**

Sufficient demonstration on hardware

**Strengths And Weaknesses:**

Strengths:
1. propose a contact configuration: a binary contact mode;
2. present a framework for modeling compliant tool-environment contact interactions by learning contact feature dynamics.
3. propose a learned model architecture to capture the dynamics of contact features, trained in a supervised fashion using real world self-supervised data.
4. design and demonstrate a controller using the contact feature dynamics to realize diverse goal trajectories.

Weaknesses:
1. authors believe the line contacts model can be straightforward to extend to patch contacts by using a richer contact descriptor, but do not proof it.
2. The model is used for compliant tool-environment contact interactions, but only test on a spatula.

**Summary Of Recommendation:**

I think the idea presented in this paper is interesting and promising.  The network architecture and data collection are clear.
More experiments are needed to proof the generalization of the framework or defined strict constraints.

---

> ### Author Response · Authors · 2022-08-28
> **Response to Reviewer DDzt**
>
> **Comment 1: Authors believe the line contacts model can be straightforward to extend to patch contacts by using a richer contact descriptor, but do not prove it.**
>
> We agree that this statement is not well supported, and have removed it.
>
> **Comment 2: The model is used for compliant tool-environment contact interactions, but only test on a spatula.**
>
> Towards demonstrating more flexibility, our extended experiments demonstrate our method’s performance across multiple spatulas, with differing geometries and physical properties. While we do not address additional tools in this work, our ability to handle spatulas with differing properties is nevertheless encouraging and suggests our method can handle tools that vary.
>
> **Comment 3: More experiments are needed to proof the generalization of the framework or defined strict constraints. For line contacts model, different geometry and physical properties (e.g., mass, stiffness, and friction) tools are needed to proof the generalization of the framework.**
>
> In our extended experiments we show performance modeling several spatulas as well as demonstrating generalization to an unseen spatula. The spatulas tested vary in both geometry and in physical properties. See Fig. 5.a. for tool geometries and Sec. 5.3 and 5.4: Obstacle-Free for experimental results.
>
> **Comment 4: How about the performance on discontinuous surface, such as v-shape surface?**
>
> Our current contact feature representation is a line, which limits the space of tool-environment interactions the model can admit. A discontinuous surface could yield several distinct contacts between a tool and the environment which cannot be represented by a line. In future work, we wish to investigate more sophisticated contact geometry representations that can admit more complex tool-environment interactions.
>
> **Comment 5: How about the performance of rotating the spatula comparing to pushing?**
>
> Our method is not inherently limited to the push/pull/tilt scrapes investigated here. So long as the dataset reflects the desired motions the method should be able to model it. Our choice of scrapes to investigate stemmed primarily from convenience in automatic labeling. As we seek to relax this necessity in future work, it should be easier to admit more complex motions into the contact feature model.

---

### Official Review · Reviewer_gMWE · 2022-07-29

**Originality:** Good
**Technical Quality:** Good
**Clarity Of Presentation:** Good
**Impact:** 2

**Recommendation:**

Weak Reject: I recommend rejecting the paper, but will not argue for my recommendation if the majority of other reviewers have a different opinion.

**Summary:**

The authors proposed a learning approach for modeling tool-environment interaction which learn the dynamics from real world data.
Key of proposed method is embedding the latent spaces from the robot's sensor data and decodes the contact feature representation which consists of binary contact value, the line of contact, and an end-effector wrench.	The authors verified the scraping operation of a clay-like object using a spatula attached to the tip of an actual robot as a task involving contact between the environment and a compliance tool.
Experimental results show that the proposed method is superior in all evaluation items (Contact force error, Binary Accuracy, and Contact Line Error). Furthermore, by adding the data extension and wrench offset action proposed by the authors, the robot is able to scrape objects from the table while avoiding contact with obstacles.

**Issues:**

As mentioned in "Strengths And Weaknesses," the lack of comparative experiments with other research methods is critical. Additional experimental results are needed to show that the proposed method is effective in solving the problem.

**Quality Of The Limitations Section:**

Additional details required

**Reviewer Expertise:**

4: The reviewer is confident but not absolutely certain that the evaluation is correct

**Robotics Focus:**

Sufficient demonstration on hardware

**Strengths And Weaknesses:**

The overall writing of the paper is easy to understandable, and the attached video also helped the reader understand. Most methods using supervised learning to learn robot policies from real-world data do not consider the adequacy of the generated trajectory or contact because they give the model predictions directly as motor commands. In contrast, the authors use MMPI to account for losses to trajectory and contact predictions, and they clearly describe their specific method. Furthermore, the authors also demonstrate the effectiveness of the proposed method by conducting multiple quantitative evaluations.

One concern is that there are few comparisons with other methods. Since the authors evaluated the accuracy of the task only based on differences in the structure of their proposed model (with or without vision and data augmentation, etc..), it is better to include the results of comparisons with other studies if the authors want to claim the effectiveness of the proposed method. Many previous studies of tasks involving contact exist. [1] showed that by using imitation learning to learn the contact force with the floor surface, the robot can properly mop the floor even if the grasping position and length of the mop are different. The reviewer has some major comments. The reviewer agrees with the issue on line 21. However, as the solution shown by the proposed method is limited, it would be better to show the consideration and results of the tool-environment interaction when the grasping position of the spatula is changed. Line 49 states that a variety of trajectories are realized, but basically the robot is just moving the spatula straight. Therefore, the diversity of trajectories is not shown. In order to demonstrate the potential of the proposed method, it would be helpful to have results for varying the positions of objects and obstacles. The reviewer understood that by collecting training data with a random action policy, contact information between the tool and the spatula could be collected. As shown in Fig. 2, the reviewers also understood that the robot predicts its behavior every step based on the observed data. On the other hand, obstacles and objects are not included in the training data. How do they recognize the objects and rub them with the spatula? It is unclear how the approach trajectory to the object is learned from the random trajectory. Explaining the data flow during learning and during execution will help the reader understand. Finally, how are the Desired contact trajectories in Figure 4 calculated? Are they arbitrarily determined by human?

[1] Sakaino, Sho. "Bilateral control-based imitation learning for velocity-controlled robot." 2021 IEEE 30th International Symposium on Industrial Electronics (ISIE). IEEE, 2021.

**Summary Of Recommendation:**

The lack of comparison with other previous researches makes it difficult to understand the effectiveness of the proposed method. The authors do not clearly state why their method is better for the issues mentioned in the introduction. In other words, the reviewer believes that the issue can be solved by other methods. It would be good to revisit additional tasks to show that the proposed method is effective in solving the issues.

---

> ### Author Response · Authors · 2022-08-28
> **Response to Reviewer gMWE**
>
> **Comment 1: No comparison to baselines.**
>
> We thank the reviewer for the relevant citation, which we have added to our related work. However, as highlighted in our response to all reviewers, the proposed baseline does not handle the case of contact geometry control, and focuses on maintaining contact. Additionally, it makes fundamentally different assumptions about data provided (demonstrations vs. random data collection). We agree, however, that comparison is needed to contextualize our result. To do this, we implemented a baseline that assumes rigidity in order to recover the contact state given the tool geometry. Please see “2. Baseline” in the response to all reviewers and Sec. 5.2 for further details.
>
> **Comment 2: It would be better to show the consideration and results of the tool-environment interaction when the grasping position of the spatula is changed.**
>
> Our updated multi spatula experiments examine our method’s performance when the geometry of the tool (and thus the relative grasping position to the contact event) changes. See Fig. 5.a. for tool geometries and Sec. 5.3 and 5.4: Obstacle-Free for experimental results.
>
> **Comment 3: Line 49 states that a variety of trajectories are realized, but basically the robot is just moving the spatula straight. Therefore, the diversity of trajectories is not shown.**
>
> We clarify that the trajectories are varied due to the changes in the contact geometry. As shown in our object scraping task, changing contact geometry allows us to perform interesting tasks, avoiding scraping out certain objects while successfully scraping others. We also note that the V-scrape involves both lateral motion and vertical motion in both directions. Our method is additionally not inherently limited to these types of motions, but was simply a design choice out of convenience; so long as the dataset reflects the desired motions the method should be able to model it.
>
> **Comment 4: Obstacles and objects are not included in the training data. How do they recognize the objects and rub them with the spatula? It is unclear how the approach trajectory to the object is learned from the random trajectory. How are the Desired contact trajectories in Figure 4 calculated? Are they arbitrarily determined by human?**
>
> The objects/obstacles are not currently processed automatically by the method - right now the trajectory that avoids obstacles and scrapes the target must be specified externally. For our experiments we design them by hand. The obstacles are seen by the model as part of the pointcloud input.
>
> We highlight that part of our investigation in scraping with obstacles was to investigate if the contact feature dynamics learned without obstacles in the dataset could generalize to the case where obstacles add occlusion and novel force feedback to the system. Our experiments indicated that our model was able to successfully generalize to handle these deviations.

---

### Official Review · Reviewer_yQUC · 2022-07-29

**Originality:** Excellent
**Technical Quality:** Excellent
**Clarity Of Presentation:** Very Good
**Impact:** 4

**Recommendation:**

Strong Accept: I recommend accepting the paper and will argue for my recommendation even if other reviewers hold a different opinion.

**Summary:**

The paper presents a robot control framework for controlling contact forces at the tip of a pre-grasped tool. Remarkably, the proposed approach considers a situation where the compliance of the grasped tool can be used to change the geometry of the contact. The authors presents a method to learn a dynamic model which allows to predict the effects to the robot actions on the contact geometry and on the robot end-effector location. This dynamic model is used in an MPC framework to control contact forces in a variety of scenarios, including controlling the contact forces at the tip of a flexible spatula to scrape a target object in presence of obstacles.

**Issues:**

- ***Action specifications***. At line 93 author mention the action trajectory $\mathbf{a}_{0:T-1}$ but it isn't clear what is the action space used for the robot. Authors should make this clear specifying if they control torques, positions and/or impedance.

- ***Typo on line 88***. $g_{i}+1$ should be $g_{i+1}$.

- ***Typo on line 81***. $\mathbb{R}^{2,3}$ should be $\mathbb{R}^{2 \times 3}$.

- ***Improve notation***. Since $G = \[ g_1, \dots, g_i, \dots, g_L\]$ are trajectories in time I suggest the notation $G = \[ g_1, \dots, g_t, \dots, g_T\]$.

**Quality Of The Limitations Section:**

Limitations are addressed clearly

**Reviewer Expertise:**

5: The reviewer is absolutely certain that the evaluation is correct and very familiar with the relevant literature

**Robotics Focus:**

Sufficient demonstration on hardware

**Strengths And Weaknesses:**

The paper presents a novel solution to a novel problem. Controlling tool contact forces is a very interesting problem which (to my knowledge) has been addressed only in the rigid case. The submitted paper considers the non-rigid case which is a significant novelty. The major limitation of the paper is the need of a complicated data-gathering phase which requires precision equipment (Photoneo) and human supervision.

**Summary Of Recommendation:**

The submitted paper presents a novel task and an interesting solution. The paper is well written and easy to follow. Videos are interesting to watch and the proposed solution is sound and technically correct.

---

> ### Author Response · Authors · 2022-08-28
> **Response to Reviewer yQUC**
>
> **Comment 1: Unclear action specification.**
>
> Thank you for identifying that, we have explicitly stated (Sec. 4.1, line 97) that we use change in end effector pose as our action space. Specifically, we represent changes in pose as a translation and an axis-angle rotation.
>
> **Comment 2: Trajectory time notation.**
>
> We use L in place of T here to emphasize that the length of the desired contact trajectory may not be the same as the horizon used for planning. We believe it is more clear to use T to indicate the time horizon of the planner.
>
> **Comment 3: The major limitation of the paper is the need for a complicated data-gathering phase which requires precision equipment (Photoneo) and human supervision.**
>
> We wish to emphasize that there is no manual human labeling, beyond inspection used to determine the heuristic that automatically labels from the high precision scans. We agree, however, that this remains potentially laborious, and may not scale to more complex contact-rich scenarios.
>
> To relieve this reliance on labels, we are interested in exploring zero-shot/few-shot learning techniques in future work to extend labels to new scenarios. Our experiments with the unseen spatula and with obstacles are early explorations of this where we showed that labels in the simple, easier to label environment can be extended to more complex scenarios where the task has changed (e.g., new spatula) or collecting new data is challenging (e.g., visual occlusions from scraping object prevent contact labeling). Another method for relieving our dependence on the data collection is to try to supervise contact indirectly. In some cases it may be easier to detect changes in the environment (such as the motion of an object and the forces experienced by the robot) which can indirectly inform the contact state of the scene.

---

### Official Review · Reviewer_r4to · 2022-08-07

**Originality:** Good
**Technical Quality:** Very Good
**Clarity Of Presentation:** Very Good
**Impact:** 3

**Recommendation:**

Weak Accept: I recommend accepting the paper, but will not argue for my recommendation if the majority of other reviewers have a different opinion.

**Summary:**

This paper presents a model-based contact servoing framework to perform control the contact between a compliant tool and the robot's environmet. Contact is parameterized as a binary contact flag, line of contact and an end-effector wrench. Dynamics are learned in a latent space with an encoder-decoder framework that is used to predict the contact parameters at every step using a pointcloud observation and the input wrench measured at the robot's wrist.

**Issues:**

- It would be nice to see experiments and analysis of generalization, both in terms of the generalization of the model to other tools, environments, surfaces etc as well as the generalization of the method to more general, parameterizable contact shapes.
- One baseline that would be interesting to compare against, especially in the cases of tracing contact areas where the tool is NOT tilted, would be to apply a larger force than required and using just vision to track the line. While this would not be able to handle partial contacts, it would be interesting to see if this method can solve the non-partial contact servoing tasks (like the straight area) as well as the proposed method.

**Quality Of The Limitations Section:**

Limitations are addressed clearly

**Reviewer Expertise:**

4: The reviewer is confident but not absolutely certain that the evaluation is correct

**Robotics Focus:**

Sufficient demonstration on hardware

**Strengths And Weaknesses:**

Strengths:
- Data collection procedure is self-supervised without needing human intervention or labelling.
- In-depth analysis of different ablations of the model to study and evaluate the effectiveness of different components of the system

Weaknesses:
- Generalization: Experiments are restricted to a single spatula. Furthermore, the surface in contact with the spatula also remains the same. Would be interesting to see generalization to new spatulas, or even just train/test on more tools as well as analyzing variations in other parameters such as the contact surface.

**Summary Of Recommendation:**

While it would be interesting to see the performance of the proposed framework in noisier, more realistic settings, the method is promising and the presented analysis is sound and well-executed. The paper is able to demonstrate controlled extrinsic dexterity with a deformable spatula while using an impedance controller, which is a promising direction as we start moving towards robots operating in human environments.

---

> ### Author Response · Authors · 2022-08-28
> **Response to Reviewer r4to**
>
> **Comment 1: Experiments and analysis of generalization (by way of other tools, environments, surfaces) and generalization of method to more general, parameterizable contact shapes.**
>
> We agree that our initial experiments did not sufficiently examine generalization. As such, we have expanded our experiments. First, we switch from modeling a single tool to multiple tools. In particular, we train our method on data collected from three different spatulas (Fig. 5.a.). We test on new contact trajectories collected with those three spatulas as well as contact trajectories on an unseen spatula, not seen by the model during training. We examine our model performance and planning performance and see that our proposed method does indeed generalize well, even to an unseen tool (See Sec. 5.3, 5.4: Obstacle-Free, and App. C.2).
>
> We agree that further experiments on varying surfaces and environments would be interesting. Given the short duration of the rebuttal period, we did not have enough time to vary the surface and environments and leave this interesting ablation to future work.
>
>
> We are additionally interested in examining our methods' performance for general, parameterizable contact shapes, though the form of that representation and how to supervise or learn the representation remains an open research question.
>
> **Comment 2: Compare our method to a planner that seeks to maintain contact throughout by ensuring a high force throughout.**
>
> We thank the reviewer for the suggestion of a baseline to compare against. As pointed out by the reviewer, this proposed baseline doesn’t handle the case of partial contacts, which we consider important to our problem formulation. We wish for a baseline that investigates the importance of modeling the tool deformations and contact in the proposed manner. As such, we implemented a baseline that assumes rigidity in order to recover the contact state given the tool geometry. Please see “2. Baseline” in the response to all reviewers and Sec. 5.2 for further details.

---

### Author Response · Authors · 2022-08-28
**Response to All Reviewers**

We thank all reviewers for their insightful comments and feedback.

We have included an updated manuscript with changes highlighted in blue. All figures and sections referenced here are with regards to the updated draft.

Here we seek to address the two major issues expressed by multiple reviewers, specifically, a lack of generalization experiments and a lack of baselines.

**1. Generalization:**
The first issue highlighted by the reviewers was a lack of experiments showing the generalization capability of our method. We address this by adding new experiments that examine our method’s performance when training/testing on data from multiple spatulas, as well as when testing on data from an unseen spatula (i.e., a spatula not seen during training). Our results indicate that our proposed method scales to handling data from multiple tools, both in model accuracy and planning performance (See Fig. 3/Sec. 5.3 and Fig. 5b/Sec. 5.4: Obstacle-Free). Additionally our method retains high dynamics modeling performance on the unseen tool and the controller applied to the unseen tool is able to realize goal contact trajectories nearly as accurately as its performance on a training tool (See Fig. C.2/Appendix C.2 and Fig. 5c/Sec. 5.4: Obstacle-Free).

**2. Baselines:**
The second issue highlighted by the reviewers was a lack of comparison to related work. We wish to emphasize that the novelty of our problem means that, to the authors’ knowledge, there are no methods that solve the same extrinsic contact servoing problem as formulated here. Reviewer r4to proposed a baseline method that seeks to maintain high force with the table to remain in contact but doesn’t address contacts on only part of the spatula edge, which we consider important to our problem statement. Reviewer gMWE suggested an imitation learning approach that learns contact forces to enable mopping, but the method also doesn’t represent or control contact geometries. We propose a new baseline that assumes rigidity in the system and assumes access to tool geometry and table location in order to compute the intersection to calculate the contact state. For full details, please see Sec. 5.2. We compare this baseline both on its predictive performance on the dataset as well as its planning performance. Our results indicate that our proposed method outperforms the method in terms of model performance (See Fig. 3/Sec. 5.3 and Fig. C.2/App. C.2) and planning performance (Fig 5b and 5c/Sec 5.3: Obstacle-Free).

We also note that for the new multi-spatula dataset and the updated obstacle-free scraping experiments, we adjusted how we label binary contact. Our originally submitted labeling technique used wrench feedback but was noisy and thus made it difficult to cleanly separate in and out of contact data, and meant even when the spatula was in contact, it might not be properly labeled. We update the binary contact labeling to be based solely on the Photoneo scan of the scene and whether we heuristically see contact occurring. This update is used in the new experiments in Sec. 5.3 and Sec 5.4: Obstacle Free. The new labeling accepts more states as in contact, that were formerly labeled as out of contact due to a) low force feedbacks while in contact, and b) noise in the sensing. This explains why our new obstacle-free experiments show higher average IoU than the original submission. In the original submission, the bottleneck for performance was more often whether the system was labeled in contact, which may not happen even if the system was in contact. Now, we believe the score is more aligned with the ability of the controller to realize desired contact geometries.

---

### Meta-Review · Area_Chair_c39B · 2022-08-10

**Recommendation:** Accept (Poster)
**Confidence:** 4

**Metareview:**

This paper presents an architecture for modeling the interaction of a compliant tool and the environment. The contact dynamics is represented by the contact indicator, contact line, and end-effector wrench. The method is demonstrated on hardware using a compliant spatula to scrape an obstacle.

The authors successfully addressed many of the reviewers concerns, most notably by adding more examples with multiple and unseen tools to demonstrate the generalization capability. The paper makes a valuable contribution to the conference by providing a novel solution to an interesting problem.


**Best Paper Nomination:**

No

---

> ### Author Response · Authors · 2022-08-28
> **Response to Area Chair c39B**
>
> As highlighted in the response to all reviewers, we have addressed the two highlighted issues in the following ways.
>
> We demonstrate generalization by our multi-spatula modeling experiments, where we show that our method can accurately represent the contact feature dynamics of multiple spatulas, as well as extend to an unseen spatula (see “1. Generalization” in response to all reviewers).
>
> We add a rigid body baseline to better contextualize the performance of our method, where we show that our learned method outperforms the baseline in terms of a) contact feature dynamics accuracy and b) extrinsic contact servoing performance (see “2. Baseline” in response to all reviewers).